# Excellent Uniformity and Properties of Micro-Meter Thick Lead Zirconate Titanate Coatings with Rapid Thermal Annealing

**DOI:** 10.3390/ma16083185

**Published:** 2023-04-18

**Authors:** Youcao Ma, Jian Song, Yuyao Zhao, Kiyotaka Tanaka, Shijunbo Wu, Chao Dong, Xubo Wang, Isaku Kanno, Jun Ouyang, Jia Zhou, Yue Liu

**Affiliations:** 1State Key Laboratory of ASIC and System, School of Microelectronics, Fudan University, Shanghai 200433, China; 2State Key Laboratory of Metal Matrix Composites, School of Materials Science and Engineering, Shanghai Jiao Tong University, Shanghai 200240, China; 3Department of Mechanical Engineering, Kobe University, Kobe 657-8501, Japan; 4School of Chemistry and Chemical Engineering, Qilu University of Technology, Jinan 250353, China; ouyang@qlu.edu.cn

**Keywords:** lead zirconate titanate (PZT), micro-electronic-mechanical system (MEMS), piezoelectric properties, dielectric properties, rapid thermal annealing

## Abstract

Lead zirconate titanate (PZT) films have shown great potential in piezoelectric micro-electronic-mechanical system (piezo-MEMS) owing to their strong piezoelectric response. However, the fabrication of PZT films on wafer-level suffers with achieving excellent uniformity and properties. Here, we successfully prepared perovskite PZT films with similar epitaxial multilayered structure and crystallographic orientation on 3-inch silicon wafers, by introducing a rapid thermal annealing (RTA) process. Compared to films without RTA treatment, these films exhibit (001) crystallographic orientation at certain composition that expecting morphotropic phase boundary. Furthermore, dielectric, ferroelectric and piezoelectric properties on different positions only fluctuate within 5%. The relatively dielectric constant, loss, remnant polarization and transverse piezoelectric coefficient are 850, 0.1, 38 μC/cm^2^ and −10 C/m^2^, respectively. Both uniformity and properties have reached the requirement for the design and fabrication of piezo-MEMS devices. This broadens the design and fabrication criteria for piezo-MEMS, particularly for piezoelectric micromachined ultrasonic transducers.

## 1. Introduction

Owing to their satisfactory piezoelectric and dielectric properties as well as high Curie temperature compared to other piezoelectric films, lead zirconate titanate (PZT) films have been widely employed in piezoelectric microelectron-mechanical systems (piezo-MEMS), such as sensors, actuators and energy-harvesters [1,2,3]. However, except above properties, the development of PZT-based piezo-MEMS devices also includes device performances optimization. For example, the configuration design of piezoelectric devices could improve performances including bandwidth, membrane displacement, output pressure/voltage, electro-mechanical coupling performance, etc. [4,5,6]. Among them, the design of piezoelectric micromachined ultrasonic transducers (PMUT) considers piezoelectric constant as an important parameter. Specifically, the dynamic displacement d_s_ of PMUT can be greatly influenced by piezoelectric coefficient e_31,f_, which could be expressed as [7]:ds=−r2·e31,f(ts+tm+tp2−zn)·Ip(r)D·Id
where r is the PMUT radius, t_s_, t_m_, t_p_ is the thickness of the substrate, bottom electrode and piezoelectric films, respectively. Z_n_ is the distance from the middle of the piezoelectric films to neutral axis. I_p(r)_ and I_d_ are integrals related to the piezoelectric bending moment and modal stiffness of the PMUT, respectively. In addition, both the sensing sensitivity G_s_ and actuation sensitivity G_t_ could be affected by [8]:$$G_{s}\propto e_{31,f}$$
Gt∝e31,fε33

Dielectric constants ε_33_ is inversely proportional to actuation sensitivity while piezoelectric constants e_31,f_ is proportional to sensing sensitivity and actuation sensitivity. Both |e_31,f_| and ε_33_ are essential for enhancing ultrasonic receiving and transmitting performance. It is worth mentioning that PZT films possess much higher piezoelectric properties than other films, such as AlN, ZnO and so on [9]. Since, |e_31,f_| is generally positively correlated with ε_33_, strategies that enhance |e_31,f_| of PZT films are of great importance [10].

In general, excellent piezoelectric properties of PZT films require perovskite crystallographic structure with epitaxial (001) orientation [11,12]. Crystal dynamics theory suggests that, among possible PZT phases, both amorphous state and metastable pyrochlore state possess higher energy than stable perovskite state [13]. However, PZT films usually manifest an amorphous structure when deposited at relatively low temperature. Thus, subsequent annealing at high temperature is necessary to realize a uniform crystallographic structure and the formation of stable perovskite phases. Thermal annealing process normally comprises two categories: (1) conventional furnace annealing (CFA) with slow heating rate, and (2) rapid thermal annealing (RTA) with high heating rate. Huang et al. [14] compared the nucleation processes and found that PZT films during RTA would hetero-nucleated on PtPb intermetallic phases at the interfaces while PZT films during CFA preferentially homo-nucleated around defects and impurities. Hu et al. [15] systematically compared the crystallization behavior of PZT films using these two strategies and concluded that the crystallization temperature can be greatly reduced using RTA process. Wang et al. [16] suggested that the crystallinity and texture behaviors can be greatly improved by controlling RTA parameters. Besides, Lu et al. [17] studied the effects of RTA parameters and found that nucleation and growth process were more sensitive to heating rate than holding time. Wan et al. [18] confirmed the feasibility to control orientations of PZT films by tailoring RTA temperature. Yamauchi et al. [19] systematically investigated as-deposited, CFA-treated and RTA-treated PZT films, and found that the crystallinity of RTA-treated sample had not only smooth surface but was also tenfold stronger diffraction intensity than CFA-treated counterpart. Velu et al. [20] fabricated PZT films by conventional annealing and RTA. Experimental results showed that RTA was beneficial to (100) orientations while conventional annealing could promote (110) orientations. However, to our best knowledge, there is a lack of evidence on understanding the correlation between the RTA parameters and the uniformity of various properties of the films on the whole wafer, which is one of the key issues in the device level fabrication of piezo-MEMS devices whatever process were underwent [21].

In this work, medium temperature (500 °C) magnetron sputtering was employed to fabricate PZT films on a 3-inch silicon wafer, and subsequent rapid thermal annealing (RTA) was performed, in order to realize perovskite structure of PZT films. Then, the uniformity at different positions on the 3-inch wafer was examined to understand the influences of RTA treatment. This “two-step” thermal treatment process successfully achieved micro-meter thick, (001)-oriented perovskite PZT films with enhanced uniform piezoelectric, dielectric and ferroelectric properties on 3-inch silicon wafers, with fluctuation <5%. The relatively dielectric constant, loss, remnant polarization and transverse piezoelectric coefficient are 850, 0.1, 38 μC/cm^2^ and −10 C/m^2^, respectively.

## 2. Materials and Methods

Pb(Zr_0_._52_Ti_0_._48_)O_3_ ceramic target and 3-inch platinized silicon substrate were employed for magnetron sputtering. The substrate was cleaned before placing in the chamber, and the vacuum chamber was then pumped to ~10^−5^ Pa. PZT films were fabricated with a thickness of 1μm, which can be controlled by deposition time. During the sputtering process, gas pressure and temperature were fixed at 0.5 Pa and 500 °C, respectively. Moreover, the target power was set at 150 W. Subsequent RTA treatment at 700 °C for 5 min was applied. Crystallographic orientation and phase structure of PZT films were characterized by X-ray diffraction (D8, Bruker, Bremen, Germany). Surface morphology of PZT films were analyzed by scanning electron microscope (SEM, Mira3, Tescan, Brno, Czech Republic). The dielectric constant of PZT films were measured by a LCR meter (TH 2838, Tonghui, China). The room temperature ferroelectric hysteresis loops (P-E) were measured by using a Radiant Precision Premium II ferroelectric tester (Radiant Technology, Albuquerque, NM, USA). Lastly, the longitudinal piezoelectric characteristics and surface morphology were characterized via Atomic Force Microscopy with piezoelectric module (MFP-3D, Oxford Instruments, Concord, MA, USA) while transverse e_31,f_ piezoelectric measurement were performed by laser Doppler vibrometer (OFV-5000, Polytec, Waldbronn, Germany). For the e_31,f_ test, PZT film cantilevers were diced from the Silicon wafer with the size of 20 mm × 2 mm × 0.5 mm, which was different from other tests of PZT film capacitors with small lateral size (generally not more than 0.2 mm in diameter). During the test, the PZT cantilevers were fixed at one end and driven by an electric voltage at the other end [22,23].

## 3. Results and Discussion

Figure 1a shows the XRD patterns of PZT films in-suit and through rapid thermal annealing (RTA) process. It could be observed that there were only pyrochlore (Py) phases in as-grown PZT films. This is due to the fact that amorphous PZT phases can transform to Py phases as low as 350 °C while Pe phases only form at temperatures higher than 525 °C [24]. However, Py phase is centrosymmetric and does not display desirable piezoelectric properties compared to Pe phase. It is well known that RTA process could significantly promote the perovskite nucleation for sputtered PZT films [25]. As shown in Figure 1a, after the RTA process, the PZT films displayed dominant (001)-oriented perovskite phases. This indicates that almost all Py phases converted into perovskite phases in these PZT films after RTA process. The existence of Py phases after RTA may be attributed to the loss of PbO, which stabilized the Pb-deficient Py phases and prevented complete phase transformation [24]. In Figure 1b,c, the cross-sectional SEM images show 1μm-thick PZT films, as well as closely packed columnar grains. Moreover, nearly no visible cracks or pores are observed after RTA process. Combining the results of XRD and SEM, introducing RTA can improve the film quality and replace the Py phases with (001)-preferred orientation Pe phases, which is crucial for achieving a good piezoelectric performance.

In practical applications, films integrated on a wafer are required to possess good uniformity in terms of microstructure and properties. Figure 1d exhibits the XRD patterns of PZT films after RTA at different locations on a 3-inch Si wafer. The inset of Figure 1d is the picture of Si wafer coated with PZT films where the three probed positions were marked. It could be observed that PZT films at different locations on the 3-inch wafer possessed similar (001) crystalline orientations. Figure 1e–g are the surface SEM images obtained from the marked positions 1, 2 and 3, respectively. PZT films at the three positions all display smooth morphology with no visible bulges or cracks. Furthermore, representative surface element distributions of PZT films at red rectangle area in position 2 are presented in Figure 1h. The Pb, Zr, Ti and O elements are all uniformly distributed across the film surface, and similar distributions are observed at positions 1 and 3. Through a quantitative EDS analysis, the composition of PZT films is confirmed to be stoichiometric with a Zr/Ti ratio of 52/48 (morphotropic phase boundary). Thus, we can conclude that PZT films deposited at different positions on the 3-inch silicon wafer possess similar microstructure, including the phase composition, crystalline orientations and surface morphology.

Dielectric, ferroelectric and piezoelectric properties of RTA treated PZT films were examined, with special emphasis on the property variation at different positions over the whole wafer. Before that, surface morphology was evaluated to guarantee the reliability of electrical characterization, as shown in Figure 2a. It could be observed that the roughness is ultra-low, 1000 nm ± 10.5 nm. This indicates that PZT films possess smooth surface without obvious defects and surface morphology does not have unfavorable effects. Figure 2b implies the relations of relative dielectric constant ε_r_ and applied electric voltage. There are typical ferroelectric butterfly loops that exhibit good repeatability for the measurements at different positions. The two peaks in the loop indicate the polarization switching, which proves the excellent ferroelectricity. The values of ε_r_ fluctuate between 500 and 1100 according to the voltage. The remnant ε_r_ was slightly shifted and had a value of about 850. This small shift indicates the existence of build-in electric field [26]. Similar to dielectric constant ε_r_, the dielectric loss tan δ in Figure 2c also varies with the voltage between 0.05–0.14. The loss in different positions fluctuates in a small range and the remnant tan δ possessed a value below 0.1. Remnant polarization P_r_, presented in Figure 2d at positions 1, 2 and 3, were 34.8, 38.1 and 38.3 μC/cm^2^, respectively. Moreover, coercive field E_c_ in different positions were nearly repeatable with zero-polarization voltage at −45.5 kV/cm and +60.5 kV/cm, respectively. The asymmetric loops could also be explained by the existence of build-in electric field and space charge accumulated at PZT films-electrode interface [27].

Figure 3a shows the variation of switching angle and piezo-response amplitude as a function of bias-voltage. There is an obvious rectangular-shape loop, indicating the successful 180° domain switching. The domain switching indicates the existence of ferroelectricity and piezoelectricity with variable piezoresponse amplitude manifesting regular butterfly-shape. Figure 3b demonstrates the voltage-dependence of transverse piezoelectric coefficients e_31,f_ of PZT films at different positions on the 3-inch silicon wafer. The inset in Figure 3b shows the configuration of e_31,f_ measurement. Transverse piezoelectric coefficients e_31,f_ could be calculated as follows [28]: (1)|e31,f|=δouths2Es3(1−υs)L2V
where h_s_, E_s_, ν_s_, L, V and δ_out_ are the thickness, Young’s modulus and Poisson’s ratio of the substrate, length of the PZT unimorph cantilever, input AC voltage, and tip displacement of free-end, respectively. The frequency of the applied ac voltage was 310 Hz, much lower than the resonant frequency of the cantilever. h_s_ = 550 μm, E_s_ = 169 GPa and ν_s_ = 0.064 for a Si substrate were used to calculate e_31,f_ [22]. With increasing voltage, the values of e_31,f_ first rose and then saturated after a threshold voltage (about 10 V). This may be attributed to the tendency of c-domains alignment to electrical field, where the volume fraction of c-domains was increasing with increased voltage [29]. This mechanism was also demonstrated in piezoelectric ceramics [30]. Similar to relative dielectric constant ε_r_, transverse piezoelectric coefficient e_31,f_ manifest consistent values across the wafer surface, especially at saturation voltage (>7.5 V). In this case, the saturated |e_31,f_| values were stable at around 10 C/m^2^, regardless of the wafer positions. Therefore, uniformity in the dielectric, ferroelectric and piezoelectric properties could be successfully achieved.

Through the RTA process, PZT films exhibited (001) preferred orientation with less pyrochlore phase, and also presented excellent piezoelectric performance. The RTA process is an effective mothed to improve film quality. Moreover, there are some theoretical supports. Dang et al. [13] investigated the annealing process using Landau model and Langevin dynamical simulations, the RTA process can be seen as long-ranged elastic interactions and these different crystallographic phases manifested various Landau-type free energy. The high heating rates enable higher energy within RTA system than sputtering, which can supply more energy to bypass the pyrochlore formation, as shown in Figure 1a. Figure 4 demonstrates the schematics of RTA and conventional furnace annealing (CFA) process, the heat released in the relaxation process was not dispersed but aided in further nucleation and crystallization of perovskite phases. Thus, as shown in Figure 4, compared to CFA with slow heating rate (Red rectangle) with A/B/C three phase transitions, RTA may skip one phase and generate A/C or B/C two phase transitions. In this work, the initial state of deposited PZT films was not amorphous but pyrochlore at different sputtering temperature, indicating state B in Figure 4. In general, PZT films deposited at low temperature (<300 °C, green rectangle) manifest amorphous state rather than pyrochlore phases forming at higher temperature (>350 °C, medium temperature). Therefore, RTA may only need to overcome energy barrier F and facilitate phase transition from pyrochlore to perovskite phases, marked as the bule rectangle in Figure 4 [24].

In addition, RTA may also overcome energy barrier D to form perovskite phases, as indicated by green rectangle in Figure 4. This is dominated by thermal kinetics and there is nearly zero relaxation. In this case, the excess energy in amorphous phases may play an important role on forming perovskite phases. The greater energy differences provide a more powerful driving force to the nucleation of perovskite phases, and the formation of pyrochlore phases as an intermediate state can be avoided effectively. Therefore, we think RTA could also improve the crystallinity and realize the completely transformation within a short time, which is beneficial to the piezoelectric properties due to more perovskite phases and less interfacial reactions between PZT films and bottom electrode [31].

## 4. Conclusions

In summary, micro-meter thick PZT films were successfully integrated on a 3-inch silicon wafer via the combination of medium-temperature magnetron sputtering and rapid thermal annealing (RTA). Specially, the uniformity of PZT films after RTA on the whole wafer was explored. The crystallographic orientations, surface morphology and elemental composition at different positions showed nearly negligible differences. Furthermore, the relative dielectric constant ε_r_, dielectric loss tan δ, remnant polarization P_r_ and transverse piezoelectric coefficient |e_31,f_| fluctuated within 5%, which were about 850, 0.1, 38 μC/cm^2^ and 10 C/m^2^, respectively. The uniformity of microstructure and properties on the whole wafer after RTA is beneficial to subsequent device fabrication. Our PZT films integrated on the 3-inch silicon wafers will definitely broaden the multiple possibilities for silicon-based piezo-MEMS applications.

## Figures and Tables

**Figure 1 materials-16-03185-f001:**
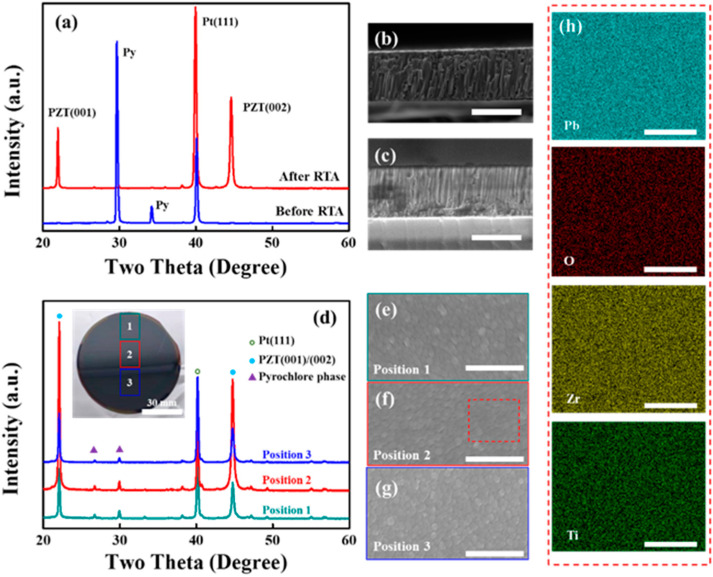
(**a**) XRD patterns of PZT films before and after RTA process; Cross-sectional SEM images of PZT films (**a**) before and (**b**) after RTA process; (**d**) XRD patterns of PZT films within different positions on a 3-inch wafer; The inserted image shows the actual position on a 3-inch wafer. SEM surface morphology of PZT thin films at (**e**) Position 1, (**f**) Position 2 and (**g**) Position 3; and (**h**) corresponding energy dispersion spectrum mapping of PZT films at red rectangle area in Position 2. All scale bars in (**b**–**f**) are 1 μm; All scale bars in (**h**) are 300 nm.

**Figure 2 materials-16-03185-f002:**
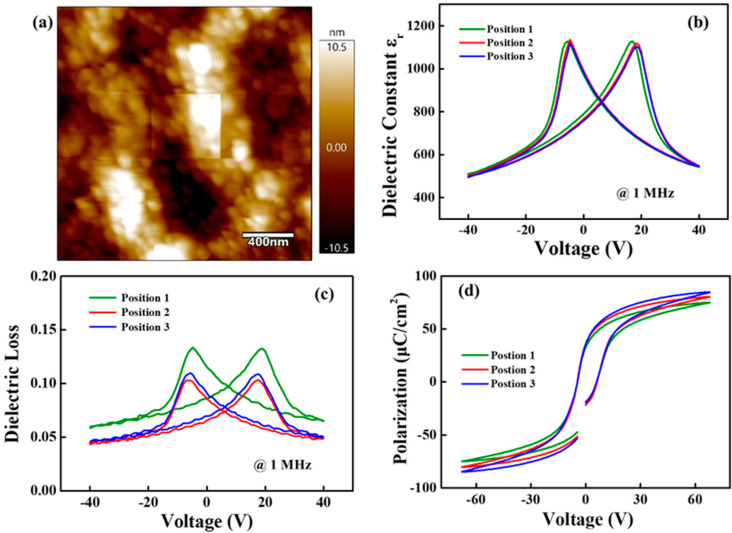
(**a**) AFM morphology of PZT films at position 2; Relations of (**b**) relative dielectric constant ε_r_, (**c**) dielectric loss tan δ, (**d**) polarization and drive voltage for PZT films at different positions on a 3-inch wafer.

**Figure 3 materials-16-03185-f003:**
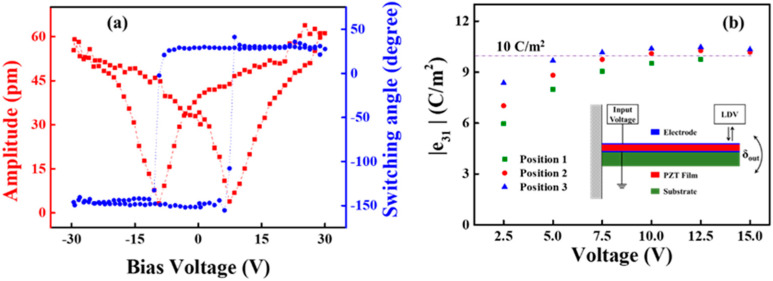
(**a**) Switching angle and piezoelectric amplitude as a function of bias voltage for PZT films; (**b**) Dependence of transverse piezoelectric constant e_31,f_ and voltage for PZT thin films at different positions on a 3-inch wafer.

**Figure 4 materials-16-03185-f004:**
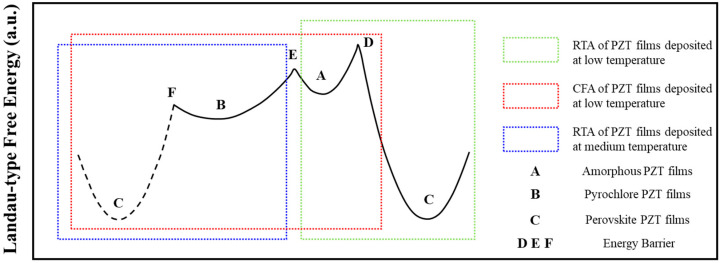
Schematics of annealing process of PZT films at different states.

## Data Availability

Data sharing is not applicable.

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
