# Peer review of "Excellent Uniformity and Properties of Micro-Meter Thick Lead Zirconate Titanate Coatings with Rapid Thermal Annealing"

_materials, 2023, doi:10.3390/ma16083185_

Round 1
Reviewer 1 Report
This paper discusses the development of a PZT layer for PMUT applications. The fabricated PZT film has been studied through XRD, SEM and EDS analysis. The film shows good performance and the effect of annealing has been discussed with detailed studies pertaining to dielectric constant, loss, remnant polarization and transverse piezoelectric coefficients. Following are the points for improvement in the manuscript 1. There are several grammatical mistakes throughout the manuscript and need thorough proofreading 2. The literature review is not comprehensive in the paper. 3. The study discusses the film characteristics of as deposited and annealed films. It would be nice to compare the results obtained for films deposited at other temperatures 100 - 500 oC and annealed at different temperatures 300 - 900 oC. A comparison of film characteristics would be interesting to study. 4. The sputtering parameters should be elaborately described. 5. XPS, AFM and FESEM/HRTEM images are desirable to verify the surface morphology and chemical composition 6. A detailed explanation of Fig. 2a will be helpful for readers 7. Again, a detailed explanation of Fig. 4 will give more insight to the material characteristics with annealing 8. How does the current film fit in the SZ model? 9. More discussions for PMUT applications may be included, how this material can improve the performance?
Author Response
Dear Editor of Materials,
We thank both reviewers for their evaluation. We have performed additional experiment to enrich the manuscript and the revisions are highlighted in yellow. Please consider for publication. Thank you very much for your attention.
Yours sincerely,
Yue Liu
Reviewer 1#
This paper discusses the development of a PZT layer for PMUT applications. The fabricated PZT film has been studied through XRD, SEM and EDS analysis. The film shows good performance and the effect of annealing has been discussed with detailed studies pertaining to dielectric constant, loss, remnant polarization and transverse piezoelectric coefficients. Following are the points for improvement in the manuscript
- There are several grammatical mistakes throughout the manuscript and need thorough proofreading
Response: We thank for the reviewer’s comments. We have carefully revised the manuscript and the revisions are highlighted in yellow.
- The literature review is not comprehensive in the paper.
Response: We thank for the reviewer’s suggestions. We have added several references in the revised manuscript.
- The study discusses the film characteristics of as deposited and annealed films. It would be nice to compare the results obtained for films deposited at other temperatures 100 - 500 °C and annealed at different temperatures 300 - 900 °C. A comparison of film characteristics would be interesting to study.
Response: We thanks for the reviewer’s comments. In fact, we have performed additional experiments, where PZT films deposited below 300°C could not achieve satisfactory crystallinity even after thermal annealing. With the increase of deposition temperature, the crystallinity of PZT films will be improved. Moreover, PZT films deposited at 500°C have no obvious change when annealed below 500°C, which indicates the same crystallinity before and after RTA process. Besides, PZT films annealed at 700°C possess smoother surface morphology. Lower or higher annealing temperature will induce cracks or other defects.
- The sputtering parameters should be elaborately described.
Response: We thank for the reviewer’s suggestions. We have added more parameters in Experiment section and the revisions are marked in yellow.
- XPS, AFM and FESEM/HRTEM images are desirable to verify the surface morphology and chemical composition
Response: We thank for the reviewer’s comments. Actually, we have performed additional experiment to characterize morphology and composition. According to the AFM test, we could observe low roughness throughout the surface, which is consistent with Figure 1 (f). Furthermore, EDS in high magnification was also employed to analyze the element distribution, which is consistent with Figure 1(h).
- A detailed explanation of Fig. 2a will be helpful for readers
Response: We thank for the reviewer’s suggestions. We have elaborated more discussions and revisions are marked in yellow.
- Again, a detailed explanation of Fig. 4 will give more insight to the material characteristics with annealing
Response: We thank for the reviewer’s comments. We have added more discussions and revisions are marked in yellow.
- How does the current film fit in the SZ model?
Response: We thank for the reviewer’s suggestions. In this work, the deposition temperature is 500°C while the melting temperature of PZT films is about 1385°C. Therefore, the ratio is 0.36. Besides, the deposition pressure is relatively low. Therefore, according to Structure Zone Model, current films fit Zone II where obvious columnar grains could observed as shown in Figure 1(b) and (c).
- More discussions for PMUT applications may be included, how this material can improve the performance?
Response: We thank for the reviewer’s comments. We have added description of PMUT application in the Introduction section.

Reviewer 2 Report
The manuscript presents a characterization based work on the lead zirconate titanate coating made by fast annealing. Due to the rapid annealing, the aligned grains provide uniform dielectric properties. In general, the article is well written. The reviewer recommends acceptance with some additional input.
1. If possible, the mapping of dielectric properties on the sample would be a better result with respect to 3 position measurements.
2. The crystal orientation and size are highly dependent on the heating/ cooling rates and heat flux directions. Hence, uniformity can be achieved in a certain spatial region. When the workpiece size is large, the cooling rate can not be uniform, whereas the heat flux can not be aligned. Please add some discussion regarding the proper sample size to remain the uniformity mentioned in the manuscript.
3. Some minor errors need to be corrected. For example, the legend in Figure 2.
Author Response
Dear Editor of Materials,
We thank both reviewers for their evaluation. We have performed additional experiment to enrich the manuscript and the revisions are highlighted in yellow. Please consider for publication. Thank you very much for your attention.
Yours sincerely,
Yue Liu
Reviewer 2#
The manuscript presents a characterization based work on the lead zirconate titanate coating made by fast annealing. Due to the rapid annealing, the aligned grains provide uniform dielectric properties. In general, the article is well written. The reviewer recommends acceptance with some additional input.
- If possible, the mapping of dielectric properties on the sample would be a better result with respect to 3 position measurements.
Response: We thank for the reviewer’s comments. Actually, we have measured more than 30 points to evaluate the dielectric properties of PZT films. However, the test of transverse piezoelectric constant e31 must be carried out in the form of a cantilever (20mm length *3mm width). Therefore, in order to keep consistent with transverse piezoelectric constant e31, we decide to choose part of results of dielectric properties to present in this paper.
Furthermore, we have also made some optical test to evaluate the dielectric properties in THz band. Dielectric constant ε could be calculated by refractive index n and extinction coefficient k as n2-k2. The measured refractive index n and calculated results of dielectric constant ε are shown in in the figure below.
- The crystal orientation and size are highly dependent on the heating/ cooling rates and heat flux directions. Hence, uniformity can be achieved in a certain spatial region. When the workpiece size is large, the cooling rate cannot be uniform, whereas the heat flux cannot be aligned. Please add some discussion regarding the proper sample size to remain the uniformity mentioned in the manuscript.
Response: We thanks for the reviewer’s suggestions. It is of vital importance to guarantee the uniformity of heating and cooling during deposition and annealing. In this work, 3 inch wafer was employed with the diameter of about 76mm. To remain the uniformity, bigger heater of φ200mm in thermal annealing. Besides, a holder with excellent thermal conductivity was also employed to support the substrate.
- Some minor errors need to be corrected. For example, the legend in Figure 2.
Response: We thanks for the reviewer’s comments. We have carefully revised some mistakes throughout the manuscript.

Round 2
Reviewer 1 Report
The authors have addressed major queries but XPS analysis and FESEM/HRTEM images are desirable for comprehensive material characterization.The AFM results should be incorporated in the manuscript.
Author Response
Dear Editor of Materials,
We thank reviewer #1 for his/her evaluation. We have incorporated the AFM results in the revised manuscript. Please let us know if there is any improvement needed. Thank you very much for your attention. Please consider for publication.
Yours sincerely,
Yue Liu
Response to Reviewer #1.
Comment: The authors have addressed major queries but XPS analysis and FESEM/HRTEM images are desirable for comprehensive material characterization. The AFM results should be incorporated in the manuscript.
Response: We thank for the reviewer’s comments. We have incorporated the AFM results in the revised manuscript as Figure 2a. Both figure caption and main text is highlighted in yellow.
Regarding XPS analysis and FESEM/HRTEM images request, we have provided FESEM/EDS from both cross-sectional view and plan-view, to ensure the uniform distribution of chemical composition. Additional XPS and FESEM results would not help to improve the current version. We have performed detailed HRTEM analysis for ferroelastic domain characterization of these thin films, as shown in Figure R1 (as attached). We decide not to incorporate these parts of information into this manuscript, because the major focus of this paper is to highlight the uniformity of both microstructure and properties.
